# Numerical Calculation and Analysis of Water Dump Distribution Out of the Belly Tanks of Firefighting Helicopters

**Tejun Zhou [1,2,]*, Jiazheng Lu [1], Chuanping Wu [1] and Shilong Lan [3]**

1 State Key Laboratory of Disaster Prevention and Reduction for Power Grid Transmission and Distribution Equipment, Changsha 410029, China

2 School of Electrical & Information Engineering, Changsha University of Science & Technology, Changsha 410205, China

3 School of Aeronautical Science and Engineering, Beijing University of Aeronautics and Astronautics, Beijing 100190, China

* Correspondence: zhoutejun1988@126.com; Tel.: +86-189-7310-2024

**Abstract:** Helicopters are more and more widely used for water dumping in fire extinguishing operations nowadays. Increasing attention is being paid to improving helicopter firefighting efficiency. Water distribution onto the ground from the helicopter tank is a key reference target to evaluate firefighting efficiency. Numerical simulations and calculations were carried out concerning water dumping out of the belly tank of a helicopter using the VOF (Volume of Fluent Model) model and mesh adaptation in ANSYS Fluent, and the effects of two parameters, the height of the tank above the ground and the wind speed, on the wake flow and water distribution were discussed. The results showed that for forward flight, the higher the forward flight speed, the less the average water depth on the ground. Similar results were obtained for flight height. The average water depth was one order of magnitude less than in the cases of the corresponding hovering helicopter for a given wind speed. As for hovering flight, the higher the wind speed, the less the average water depth on the ground. The simulation results were basically consistent with the conclusions of water dump tests of fire-fighting equipment carried by helicopters. For example, when the helicopter flew at a forward flight speed of 15 m/s and the tank bottom was 30 m above the ground, the area covered by the dumped water would be 337.5 m$^2$, and the average water depth accumulated per square meter would be 0.3 cm. This result was close to the 0.34 cm obtained under Hayden Biggs's test condition with a forward flight speed of 70 km/h and a height above the ground of 24 m.

**Keywords:** helicopter; tank; water dump; wind speed; numerical calculation

## 1. Introduction

Forests in China cover an area of nearly two million square kilometers, accounting for nearly 20% of the country's land area. Forest fires occur frequently every year, causing extensive damage to natural resources, the ecological environment, and production facilities [1–3], and even sometimes causing heavy casualties. Therefore, it is very important to carry out proper monitoring, early warning, and forest fire suppression. During forest fire suppression, helicopters can rapidly reach the forest, where it is difficult for people to enter to extinguish the fire. The suppression achieved by the helicopter's water dumping on the fire line and the fire head reduces the intensity of effort by ground personnel against the fire and reduces the casualty rate. At present, firefighting departments in the United States, Canada, Russia, Japan, and other countries are equipped with many firefighting helicopters to execute tasks such as forest fire suppression and high-rise building firefighting [4,5]. Because the aviation industry in China started late, it has lagged behind the developed countries just mentioned in both research and development and the quantity of firefighting helicopters. Moreover, technology and applications related to helicopter firefighting are also in need of development. Helicopter firefighting mainly includes onboard tank

(belly/under-deck) water dumping, bucket/capsule water dump firefighting, fire water-monitor firefighting, and fire water-bomb drop firefighting. The firefighting effect is closely related to the water distribution in the target region [6,7]. Studies on water distribution in onboard tanks and bucket/capsule water dump firefighting mainly came from flight tests. The following is a review of studies on water distribution in onboard tank firefighting and bucket /capsule water dump firefighting [8–15].

Experimental data from the studies just described are as follows, where the water depth accumulated was an average, whereas the data from Xie Yingmin et al. [12] were merely the results on flat ground. The results of Wu Zepeng et al. [8] show that if the bucket was more than 30 m above the fire scene, the water dumped would be atomized completely, whereas the results of both Xie Yingmin et al. [12] and Zhou Tejun et al. [13] showed that, at a height of 30 m, the water depth accumulated was still 0.1–0.2 cm. The results of Chen Zhaopeng et al. [10] and Zhou Wanshu [11] showed that at a height of 50 m, the water depth accumulated could reach 0.2 cm, and could even reach 1.25 cm under the no-wind circumstance, as proved by Chen Zhaopeng et al. [10]. These results were obviously inconsistent with each other. Such an inconsistency might have been due to several aspects, for example, whether foaming agent and other surface-active materials had been added to the water. If so, the materials could reduce the formation of small water droplets. In addition, the bucket bottom valves might have been of different sizes and shapes, which would have an impact on water flow and pattern, thus resulting in differences in water dump coverage areas and the average water depth on the ground.

Because renting helicopters is expensive, the firefighting tests involve both great material consumption and high cost, and fine data cannot be obtained through conventional test techniques. Therefore, the helicopter firefighting test is restricted; however, as one of the study objectives, numerical simulation can either supplement or even replace such tests. With current physical firefighting process knowledge of water dumping out of helicopters, the application of relevant models, and constant development of numerical simulation software and computer performance, numerical simulation of firefighting by water dumping out of helicopters has developed gradually from nothing. However, its technical level has not yet reached maturity, and there are few simulation studies of firefighting by water dumping out of helicopters. Two such studies will be briefly introduced in the following.

In the calculation model of X. Zhao et al. [16], the fact or principle used to set the diameter of the water droplets was not specified, nor were the calculation results verified by experiment. If the distribution rule of water droplet diameters were known beforehand, the calculation scheme of X. Zhao et al. would be feasible. In the study of Satoh et al. [17], the actual rotor wings of the helicopter were not simulated in the numerical simulation, and the downwash velocity field generated from the rotor wings was approximated by setting a downward air velocity of 30 m/s at the upper boundary of the computational domain, an approach that greatly reduced computational effort. Borisov et al. [18] performed a relatively complete study on water dump firefighting simulation. The blades were not simulated; instead, a downward speed was imparted to the plane of the lower blade by means of a virtual blade, in accordance with the measured velocity distribution, and a corresponding velocity was imparted to the plane of the upper blade.

The above discussion confirms that when a model with low degree of approximation was used to simulate a tank or bucket water dump process, a relatively accurate water distribution was not provided, and the accuracy of firefighting effect evaluation was affected. In addition, regarding the influence of the height of the tank or bucket above the ground on water distribution over the ground, there were also inconsistent conclusions in these articles. Evaluation of firefighting effect can be expressed by the water volume arriving at the ground fire source, where height above the ground is an important parameter influencing ground water distribution. In addition, because natural wind speed is hard to control, there were few reports in the literature concerning the influence of wind speed on water distribution.

## 2. Helicopter and Tank Models

In the present paper, simulation calculation of helicopter tank water dumping was performed using two parameters: the height of the helicopter tank bottom above the ground and the wind speed. The volume-of-fluid model (VOF model) was used to calculate the air-water interface, and dynamic mesh adaptation was used to better differentiate the air-water interface. The rule of influence of the height of the helicopter tank bottom above the ground ($H$ = 10 m, 20 m, and 30 m) and the wind speed ($U$ = 0 m/s, 5 m/s, 10 m/s, and 15 m/s) on the distribution of a water dump was given to provide theoretical guidance for a helicopter firefighting operation scheme. In this approach, to calculate the rotor wake, the acting disc theory of a helicopter was used instead of a simulated blade. This was more convenient to use than the virtual screw disk of Borisov et al. [18], which yields basically the same accuracy, and was more accurate than the jet model of Satoh et al. [17]. To calculate the water dump, this paper simulated the water discharge process of real water tank directly, while Satoh et al. [17] and Borisov et al. [18] did not simulate the water dump process. For the two-phase, water–air flow, this paper adopted the volume-fraction model for calculations, which is superior to the virtual gas model of Satoh et al. [17]. However, as this model is restricted by the extremely numerous calculations required, we did not calculate the breakup of drops; we note that the drops calculated by Borisov et al. [18] were also not real fluid drops, but water drop test particles representing many fluid drops.

An H125 helicopter equipped with an Isolair Eliminator II belly firefighting tank (https://www.fs.fed.us/t-d/pubs/html/95571307/95571307.html, accessed on 2 May 2022) was taken as the simulation prototype in this paper. Figure 1 shows a water dump out of an H125 helicopter equipped with the Isolair Eliminator II belly firefighting tank. Modeling was performed in accordance with the geometrical parameters of the H125 helicopter and the tank (ANSYS Design Modeler), neglecting some parts and configuration details such as the aero-engine case, the bracket under the fuselage, and the tail rotor. In this way, the helicopter and tank models used for simulation were obtained, as shown in Figure 2. The action of the blade on air was viewed as a pressure difference acting above and below a disc, specifically the *Fan* model in Fluent, which was more accurate than the jet flow approximation used by Satoh et al. [17]. If the geometrical parameters and operating parameters of the blade were known, the momentum source method [19] could also have been used to simulate the action of the blade on air, which would have been more accurate than the acting disc theory, and there is also a momentum source model in Fluent. Both the *Fan* model and the momentum source model eliminate the large numbers of mesh cells required to simulate actual blades, while still obtaining a relatively accurate time-mean wake flow field.

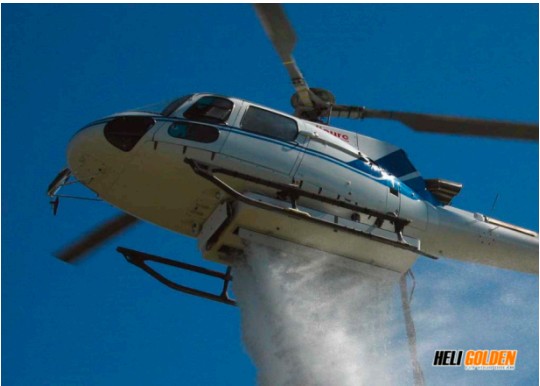

**Figure 1.** H125 firefighting helicopter and Isolair Eliminator II belly firefighting tank (from http://www.isolairinc.com/_gallery/4600-350A.jpg, accessed on 2 May 2022).

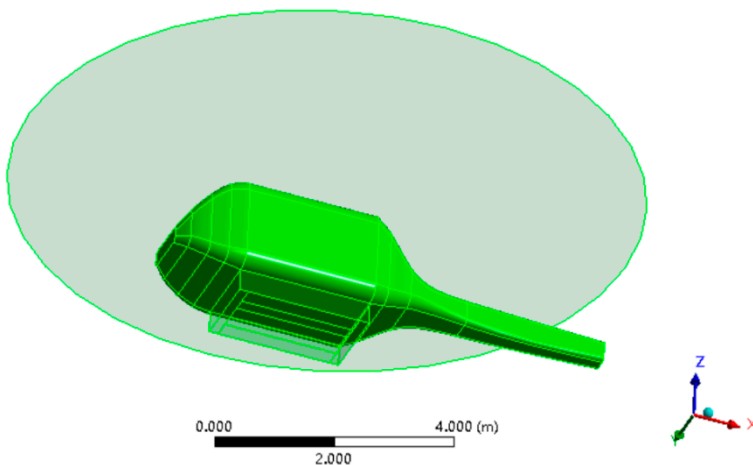

**Figure 2.** Helicopter and tank model in the simulation.

The tank used in the calculation was 2.32 m long, 1.38 m wide, and 0.4 m high, with two long narrow valves, each 2 m long and 0.23 m wide with a gap of 0.3 m. The tank had a maximum capacity of 1280.64 L. To maintain consistent air pressure inside and outside the tank at the time of water dumping, a long narrow ventilation opening was designed over the rear of the tank (see Figure 3) with dimensions of 1.38 m × 0.05 m.

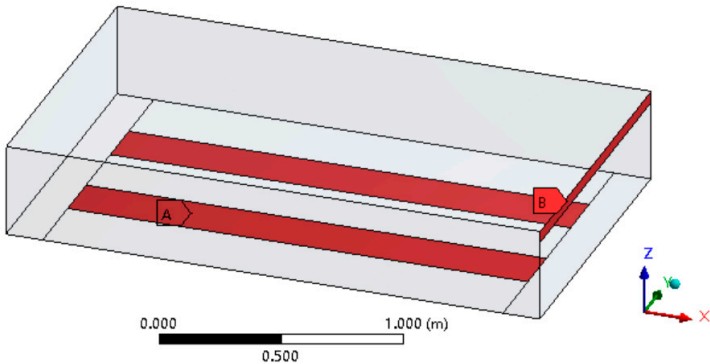

**Figure 3.** Details of the tank model (A is the water-dump valve, and B is the ventilation opening, for the dimensions of the tank and valve, see the description given in the main text).

## 3. Water Dump Simulation Model of Helicopter Belly Firefighting Tank

A parametric study was carried out in this paper regarding two key parameters that influence the distribution of the water dump: the height (H) from the tank bottom to the ground and the wind speed (U). Several examples for different heights (*H* = 10 m, 20 m, and 30 m) and wind speeds (*U* = 0 m/s, 5 m/s, 10 m/s, and 15 m/s) were calculated. The results of a calculated example are given in detail below (the height (*H*) from the tank bottom to the ground = 20 m, and the headwind speed *U* = 15 m/s), including mesh independence verification, the calculation format, and calculation scheme. This example was selected for a detailed analysis for the reason that, in this example, the height was moderate while the wind speed was relatively big, which caused very significant changes in the trajectory and shape of the water masses. Other calculated examples are given in Section 4, where the results of different heights above the ground and different wind speeds were investigated, and the rule of water distribution on the ground was generalized.

When the helicopter flew above the fire scene to prepare for a water dump, the airflow was generally stable surrounding the helicopter (the fire scene model was temporarily left out of consideration in this study). Therefore, before the water dump is calculated, the stable airflow field should be calculated first. This was then taken as the initial scenario to start transient calculations of the water dump.

### 3.1. Mesh Independence Verification

This section discusses the influence of mesh cell size on flow field results and especially on the resolution of the water–air phase interface. Two sets of meshes were used; namely, one set consisting of a basic mesh, and another with a fine grid. The verification was divided in two steps, where the first step compared the calculation results of the rotor flow field before water dump in the two sets of meshes, and the second step compared the calculation results of the two-phase, water–air flow after water dump in the two sets of meshes.

Figure 4 shows the simulation field of the helicopter dumping firefighting water. The blade disc diameter of the H125 helicopter (D) was 10.69 m, the distance from the front of the disc to the entry plane of the computational domain was 6D, the distance from the rear of the disc to the exit plane of the computational domain was 8D, the distance from the disc to both boundaries of the computational domain was 6D, and the distance from the disc to the upper boundary of the computational domain was 7D.

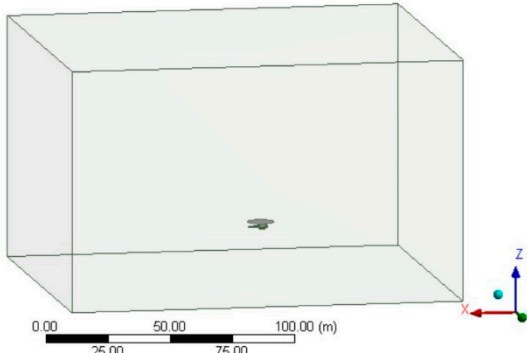

**Figure 4.** Computational domain of the helicopter.

For example, when the tank was 20 m from the ground, meshes generated by ANSYS Meshing, there were a total of 695,114 mesh cells (tetrahedrons and triangular prisms). The maximum mesh cell size inside and outside the tank was 0.05 m, the maximum size on the fuselage and on the disc of the rotor wings was 0.1 m, the maximum size on the external boundary plane of the computational domain was 5 m, and there were boundary-layer meshes (triangular-prism cells) on the disc of the rotor wings, the fuselage surface, the tank wall surface, and the ground. The thickness of the boundary layer was set to 0.05 m, there were 10 layers of mesh cells inside the boundary layer, and the mesh growth rate was 1.2; this mesh is referred to as the basic mesh in what follows. In Figure 5, the meshes near the helicopter and the tank on the longitudinal symmetry plan of the computational domain are given; the mesh cells in the tank can also be seen.

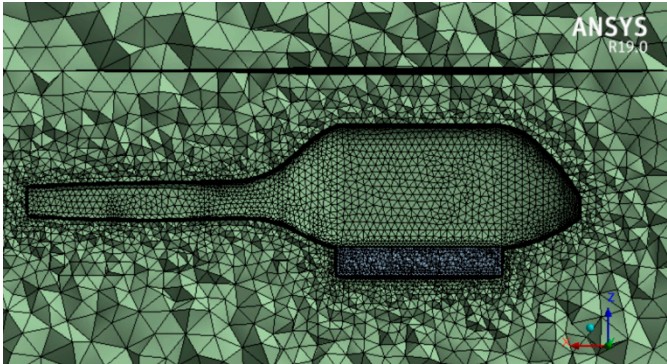

**Figure 5.** Meshes near the helicopter and tank on a longitudinal symmetric plan of the computational domain (there are boundary-layer meshes on the fuselage surface, the external tank surface, and both sides of the blade disc).

These new mesh cells are hereinafter referred to as fine meshes. They covered the spatial region from the bottom of the helicopter belly firefighting tank to the ground, which was 6.7 m long, 7 m wide, and 20 m high (as shown in the left side of Figure 6, the "body of influence" in ANSYS Workbench meshing was used), the maximum mesh cell size was set to 0.2 m. There were a total of 2,061,091 mesh cells, about three times the number (695,114) in the basic mesh cells described earlier. As will be seen later, the total number of mesh cells calculated up to 1.9 s reached 2,460,231 due to adaptive refinement of meshes.

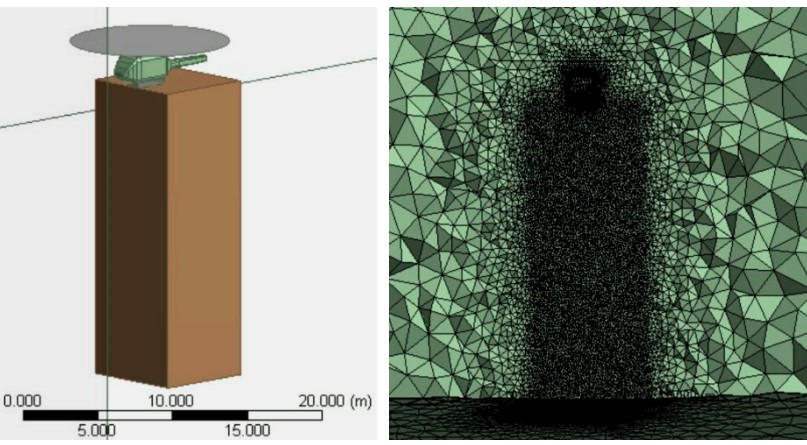

**Figure 6.** Fine mesh space region and mesh distribution.

The calculation method, format parameters, and self-adaptive mesh parameters were all set the same in the two sets of meshes. Results from the fine meshes are given below and are compared with the results of the basic meshes. Figure 7 shows the pressure distribution on the longitudinal symmetry cross section of the initial scenario as calculated with fine meshes. There was almost no difference compared with the results of the basic mesh. In fact, the place where mesh refinement was performed was the space below the helicopter belly, and compared with the pressure gradients near the helicopter rotor wings and near the fuselage surface, the pressure gradient in this space was relatively gentle, and the basic meshes were sufficient to calculate the pressure field accurately.

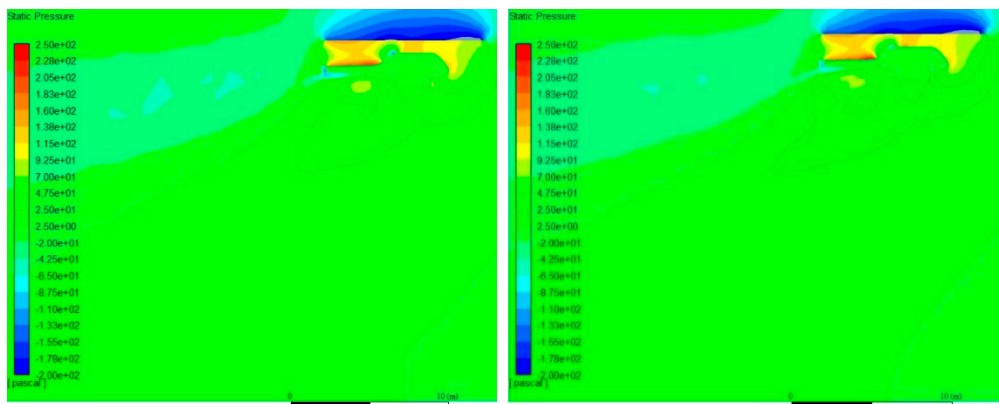

**Figure 7.** Pressure distributions on a longitudinal symmetry cross section of the initial scenario calculated with fine meshes (the ground is at the ruler in the figure, the left picture is the basic mesh, and the right picture is the fine mesh).

Next, we compared the calculation results of two-phase, gas–fluid flow in the two sets of meshes. Figure 8 gives contour surface diagrams for water–air phase volume fractions of 0.01, 0.1, and 0.5 calculated with fine meshes at $t = 1.9$ s (i.e., the water mass touched the ground just at this time). It is obvious that the fine mesh could yield a finer water–air interface and smaller water mass, but the physical quantity of interest in this paper is the

average water depth (i.e., the average water depth that is equal to the result of water-dump quantity divided by the water coverage area). The longitudinal and transverse dimensions of the water mass near the ground in Figure 8 were about 3 m × 4 m, and the calculation result of the base grid was about 4 m × 7 m, which shows that a more accurate water distribution could be obtained using the basic meshes. In Section 4.4 below, the average water depth accumulated as calculated in this paper was basically consistent with that under the corresponding condition by Xie Yingmin et al. [12], which showed that it was suitable to use the basic meshes to calculate the region covered by water and the average water depth. In addition, a third finer mesh was not taken into consideration in this paper since the calculation quantity of the fine mesh was sufficient.

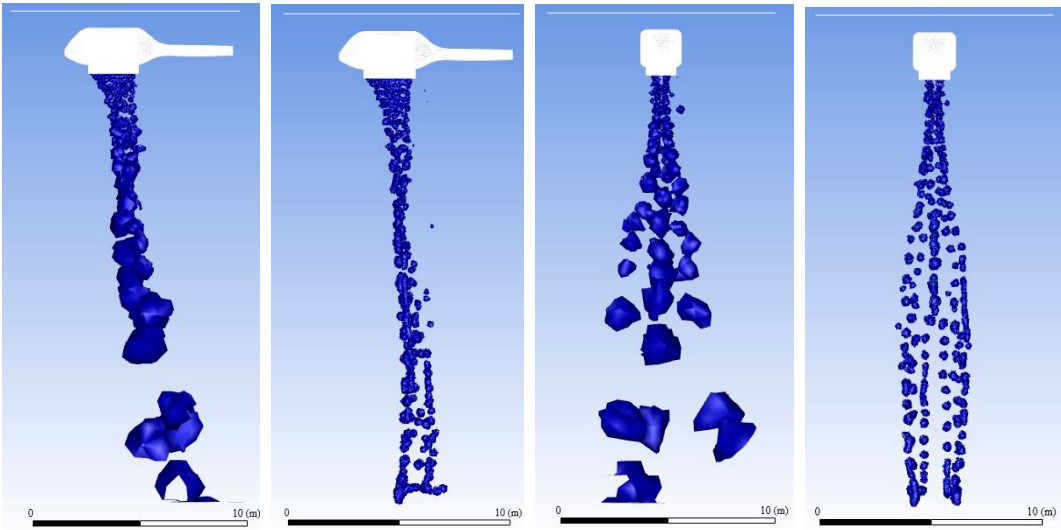

**Figure 8.** Contour surface of the side view and front view for a two-phase, water–air volume fraction of 0.01 at *t* = 1.9 s.

### 3.2. Analysis of Initial Scenario Results before Water Dump

This section first gives the calculation results of the initial flow field before the water dump. VOF can be either enabled or disabled at the time of initial scenario calculation, and VOF was disabled in this study. The internal space of the tank was set to air, and the valves and the ventilation opening were set as solid wall surfaces. The calculation was performed by a steady-state algorithm, using the k-omega SST turbulence model [20]. Since complex separation only existed near the helicopter and water tank, the wake flow with which we were concerned under the water tank was not so complexed, and to which common turbulence models were applied; hence, the equation calculation format was SIMPLE, and basic meshes were used.

Figure 9 shows the calculated residual change curve. When the number of iterations was greater than 200, the residue remained basically unchanged. The velocity field (Figure 10) and the pressure field (Figure 11) on the longitudinal symmetry plane of the fuselage at 1000 and 2000 iterations were taken, respectively, as their simulated values, and the pressure shown in Figure 11 was gauge pressure. The velocity and pressure fields at 1000 and 2000 iterations were very consistent, showing that the calculation had reached steady state. Flow field data at 2000 iterations were selected as the initial scenario for subsequent calculations of the water dump.

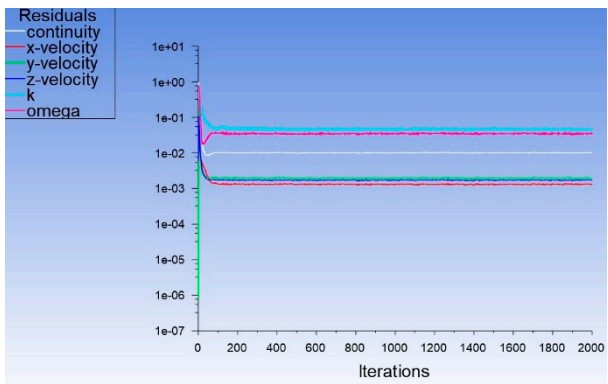

**Figure 9.** Residual change curve of the steady-state calculations.

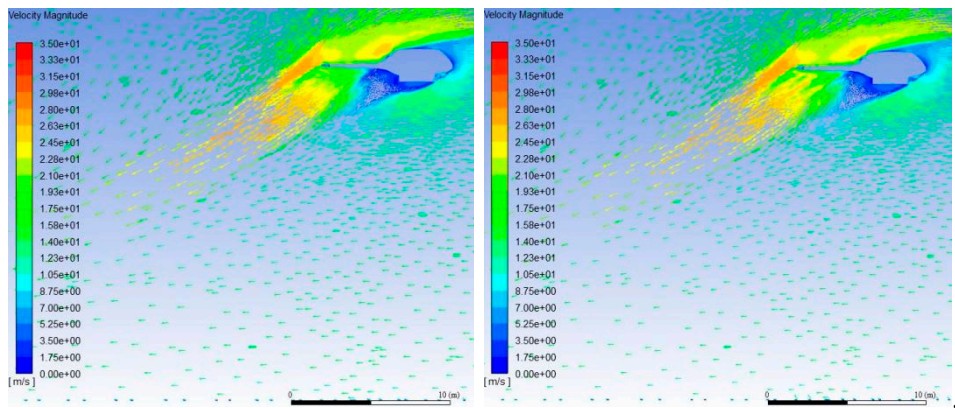

**Figure 10.** Velocity vector on the longitudinal symmetry plane of the fuselage in the initial scenario (1000 iterations on the (**left**), 2000 iterations on the (**right**)).

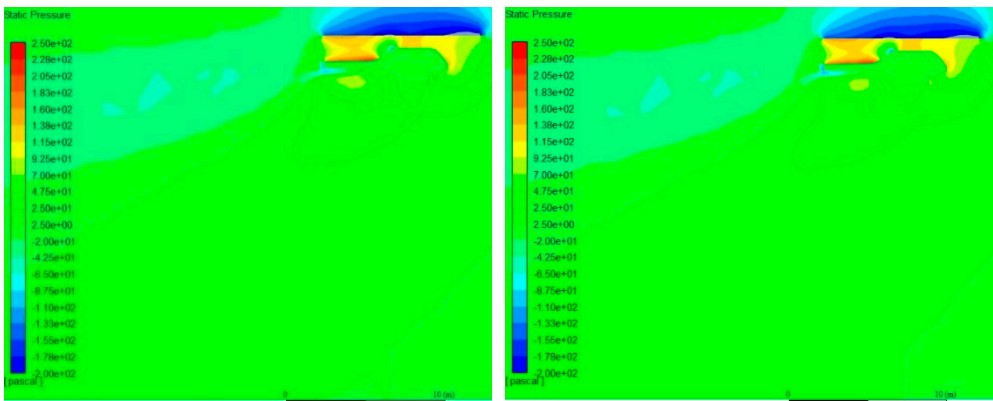

**Figure 11.** Pressure distribution on the longitudinal symmetry plane of the fuselage in the initial scenario (1000 iterations on the (**left**), 2000 iterations on the (**right**)).

### 3.3. Transient Result Analysis of Water Dump

When the transient calculations of the water dump were enabled, steady-state (1600 iterations) data were read in, and the valves and ventilation opening were set as internal boundaries. Most of the space inside the tank, 2.32 m × 1.38 m × 0.35 m (length × width × height), was defined as a region called *Region0* using cell registers in Fluent, and the initial value of this region was adjusted by *Patch* when Fluent was initialized. Specifically, the phase in this region was adjusted to water, and therefore there was initially 1.12 m$^3$ of water in the tank. Note that there was still air inside the tank at a height of 0.35–0.4 m, which connected with the air outside the tank through the ventilation opening. Data for other spaces required no

initialization and were maintained at the input steady-state values. The turbulence model and the spatial calculation format were kept the same as for the steady-state calculation.

Time was advanced by a first-order implicit method with a fixed time step of 0.001 s, corresponding to the rapidest water mass movement at approximately 1–2 cm. This ensured that there was sufficient time resolution, and the maximum number of iterations per time step was set to 50, so as to ensure residuals below $10^{-5}$. Transient calculation was enabled to $t$ = 0.01 s, and the residual change was as shown in Figure 12. It is apparent that the residuals were significantly reduced by two or three orders of magnitude after bridging the transient calculation. If the transient calculations were continued, the residual of each equation could be maintained at $10^{-5}$ to $10^{-6}$ until the calculation ended at 1–2 s (as was the case with all transient calculation examples in this paper). When $t$ = 0.01 s, the VOF distribution of water and air inside the tank and at the valve was as shown in Figure 13. If a cross section were taken in the middle of the tank vertical to the x-axis, meshes inside and below the tank could be seen (Figure 14), there were some flat triangular cells, which were formed by cutting off tetrahedral cells in cross section, and most of the mesh cells were near-regular tetrahedra (Figure 5).

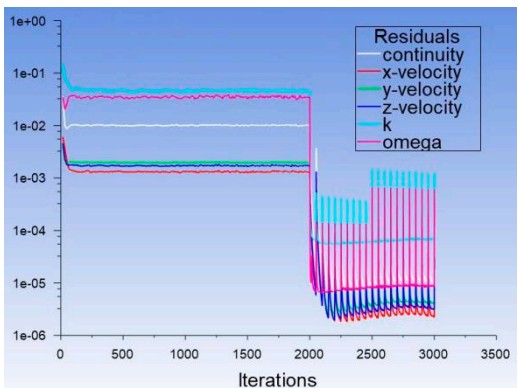

**Figure 12.** Residual change curve of steady-state calculations bridging to transient calculations (until $t$ = 0.02 s).

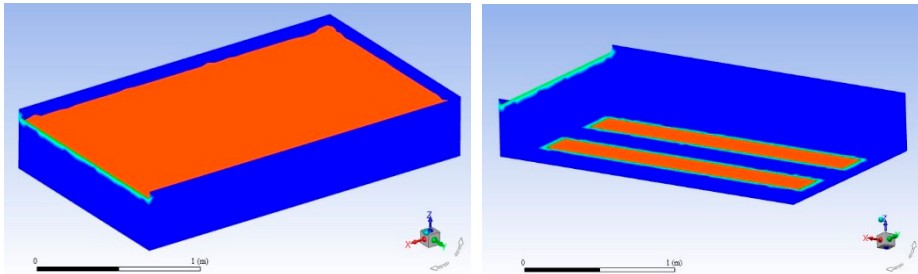

**Figure 13.** VOF distribution of water and air inside the tank (**left**) and at two values at the tank bottom (**right**) when $t$ = 0.01 s (red indicates water).

Dynamic mesh adaptation (based on the curvature of the water–air interface) was enabled after the tank water dump had been enabled for 0.01 s. Meanwhile, mesh adaptation was set at one per time step, with the time step still being 0.001 s, and for each time step, mesh adaptation, the coarsening threshold ($10^{-8}$) and the refining threshold ($10^{-2}$) were implemented. In addition, dynamic self-adaptation was enabled, the maximum level of refinement was set to 2, the minimum cell volume was set to $1.25 \times 10^{-4}$ m$^3$ (the corresponding mesh cell was 0.05 m), and mesh adaptation load balancing was used to improve parallel efficiency. The calculation lasted from $t$ = 0.01 s to $t$ = 0.45 s. The VOF distribution and the meshes of water and air on the same cross section are shown in Figure 12. The meshes near the water–air interface were encrypted to a certain extent so that the water–air interface could be better distinguished.

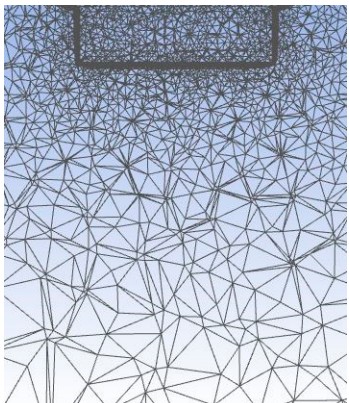

**Figure 14.** Mesh distributions inside and below the tank in cross section.

Figure 15 shows the water in the tank and nearby meshes at $t$ = 1.9 s of the water dump. At this time, most of the water in the tank has been dumped. Figure 8 shows the contour surface with a water–air volume fraction = 0.01 in space (the ground is at the ruler in the figure). The side view in Figure 8 shows that large water masses have been subjected to a certain influence by Level-3 wind, the dumped water has been deflected in the wind direction, the water mass near the ground has been deflected in the wind direction by 3–4 m, and the longitudinal dimension of the water mass has reached about 4 m. The front view in Figure 8 shows that under the action of wind, the water flow has developed laterally, and that the sideways deflection of the water mass near the ground is about 7 m. This means that the area covered by the water dump on the ground is approximately 4 m $\times$ 7 m = 28 m$^2$, under the assumption that all 1.12 m$^3$ of the water in the tank was dumped to this region. The average water depth accumulated in this region would then be approximately 0.04 m. Although in fact small water droplets would be created and would fly away with the wind, thus causing the region covered by water to be larger than the value calculated above, in the dense mesh calculation (Figure 8), the area covered by the water dump on the ground was approximately 3 m $\times$ 4 m = 12 m$^2$. Therefore, taking the two circumstances causing contrary results into full consideration, the region covered by water might not have been underestimated in the calculation using basic meshes. Hence, the region covered by water and the average water depth accumulated from the above basic mesh calculations were used as important data for drawing conclusions.

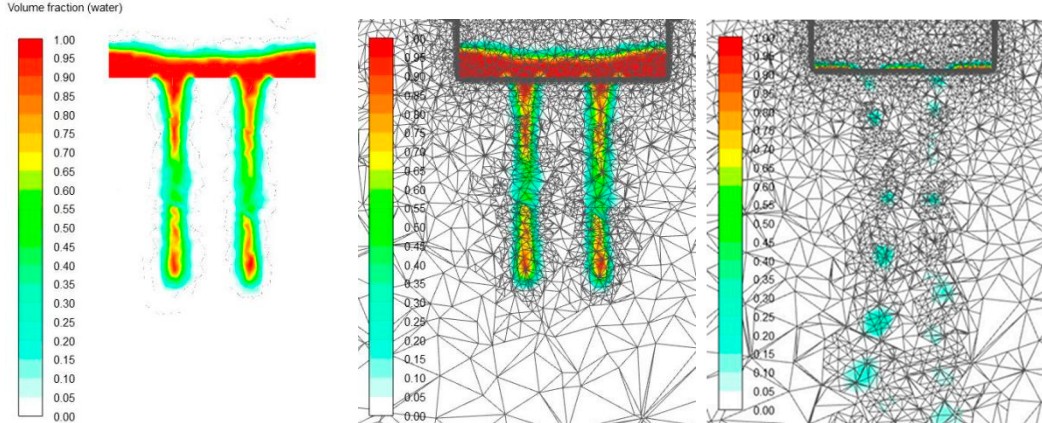

**Figure 15.** VOF distribution and network of water and air on the cross section inside and under the tank ($t$ = 0.45 s and $t$ = 1.9 s).

Comparing the side view in Figure 16 with the water dump picture in Figure 17 reveals a similar important phenomenon: the water flow was not smooth or continuous, but experienced many interruptions. Only a rough qualitative comparison could be performed

because the forward flight speed of the helicopter in the picture and the wind speed were unknown. Except for the fact that small water droplets could not be differentiated in the calculations, the water masses were of similar qualitative distribution, showing that the calculated results were qualitatively correct. Figure 18 gives a local panoramic view of the two-phase, water–air volume fraction contour surface below the helicopter belly at two moments, $t = 0.9$ s and $t = 1.9$ s. The water–air interface at $t = 0.9$ s was relatively more continuous than that at $t = 1.9$ s, because the water dump flow was greater at the former. These images resemble the photograph of the helicopter belly firefighting tank water dump in Figure 1. Because of the lack of quantitative experimental data in the literature, the results calculated here were only qualitatively compared with the actual pictures.

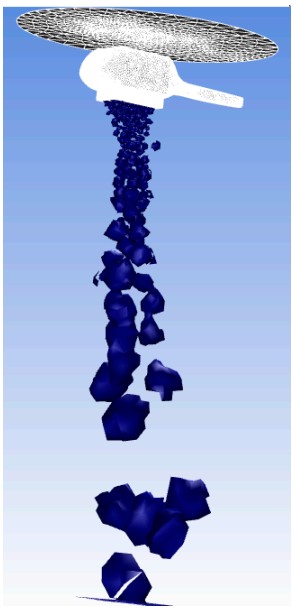

**Figure 16.** Panoramic view of the contour surface for a volume fraction = 0.01 in space at $t = 1.9$ s (the side and front views are shown in Figure 8).

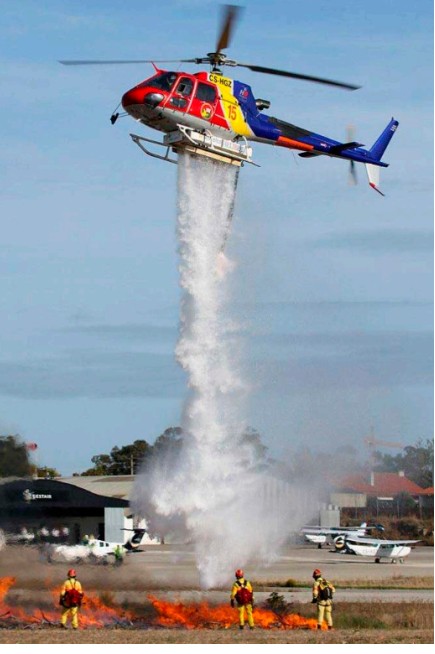

**Figure 17.** AS350B2 firefighting tank used for water dump firefighting (found online at http://www. isolairinc.com/_gallery/4600-350D.jpg, accessed on 2 September 2021).

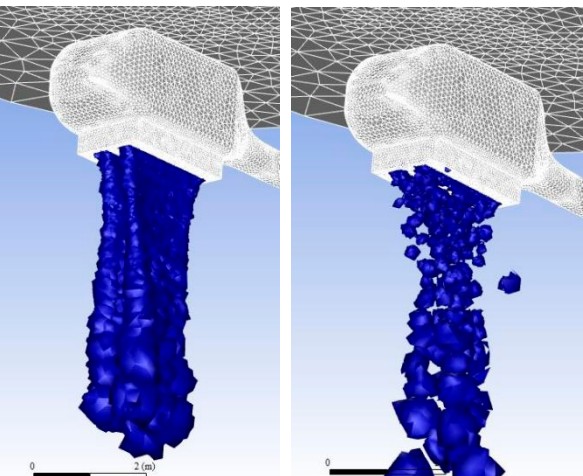

**Figure 18.** Local panorama of contour surface when the volume fraction of the water–air phase below the helicopter belly was 0.01 ((**left**), $t$ = 0.9 s; (**right**), $t$ = 1.9 s).

In fact, it is very difficult to simulate the movement of real water masses and water drops after water dump, regarding which a discussion is made below. The main force on the water mass by the airflow was in direct proportion to $\rho_{air}U^2D^2$, and the force of gravity on the water mass was in direct proportion to $\rho_{water}D^3$g, where $D$ was the equivalent diameter of the water mass, $U$ was the velocity of the water mass relative to the airflow, and g was the acceleration of gravity. When the force exerted on the water mass by the airflow was comparable to that of gravity, the water mass could significantly move along with the airflow. However, it can be inferred that only water droplets with diameter less than 1 cm could move significantly with the airflow, and the smaller the water droplets, the more closely they moved with the airflow. In our calculations, the minimum mesh cell scale was about 0.05 m, and it was impossible to distinguish water droplets less than 0.01 m in diameter. The equivalent diameters of the water masses of VOF-to-DPM [21] conversion in ANSYS Fluent were set to 0.01 m; therefore, the conversion was not activated. If a smaller mesh cell size and a smaller equivalent diameter of VOF-to-DPM water mass were adopted, the movement of small water droplets with the airflow could be calculated.

Interruption of the water–air interface occurred because of instability on the interface between the falling water flow and the air at greater Weber number. The interface then formed a complex curved surface, which was then broken into large water droplets, which would also experience instability with the air interface, leading to secondary breakage and creating still smaller water droplets. The water distribution of the water dump out of the helicopter involved a distance of dozens of meters from the tank to the ground and the passage of several seconds. The instability and breaking of the water flow and the secondary breakage of large water droplets occurred at a smaller space and time scale [22–25], which was nearly impossible to consider simultaneously in the simulation calculations. A water droplet breakage model would generally be used (many breakage models are available in ANSYS Fluent).

If sufficient calculation resources can be obtained, the mesh should be finer as much as possible, which can make the simulation of the water–air interface finer, and, if possible, one should adopt the breaking-drop model. A further analysis of the calculation results produced by the fine mesh is provided below. Figure 19 gives contour surface diagrams when the water–air phase volume fraction calculated with fine meshes at $t$ = 1.9 s was 0.01, 0.1, and 0.5. The three contour surfaces were of relatively similar form, showing that the two-phase, water–air interface was reconstructed with reasonable accuracy under this fine mesh and with the mesh adaptation calculations. Figure 20 shows a local enlargement of the results near the ground in Figure 19, with the contour surface with volume fractions of 0.01 and 0.5, and in addition shows the mesh cells at that location on the contour surface. The maximum mesh cell size of the refined meshes near the ground was 0.2 m, and the

minimum cell size of the mesh adaptation was 0.05 m. Therefore, the maximum mesh cell size was 0.05 m, which meant that water droplets smaller than 0.05 m could not be distinguished. Due to insufficient mesh refinement in the present calculations, the "VOF-to-DPM" model in ANSYS Fluent was not activated, and therefore the water droplet breakage model was not used.

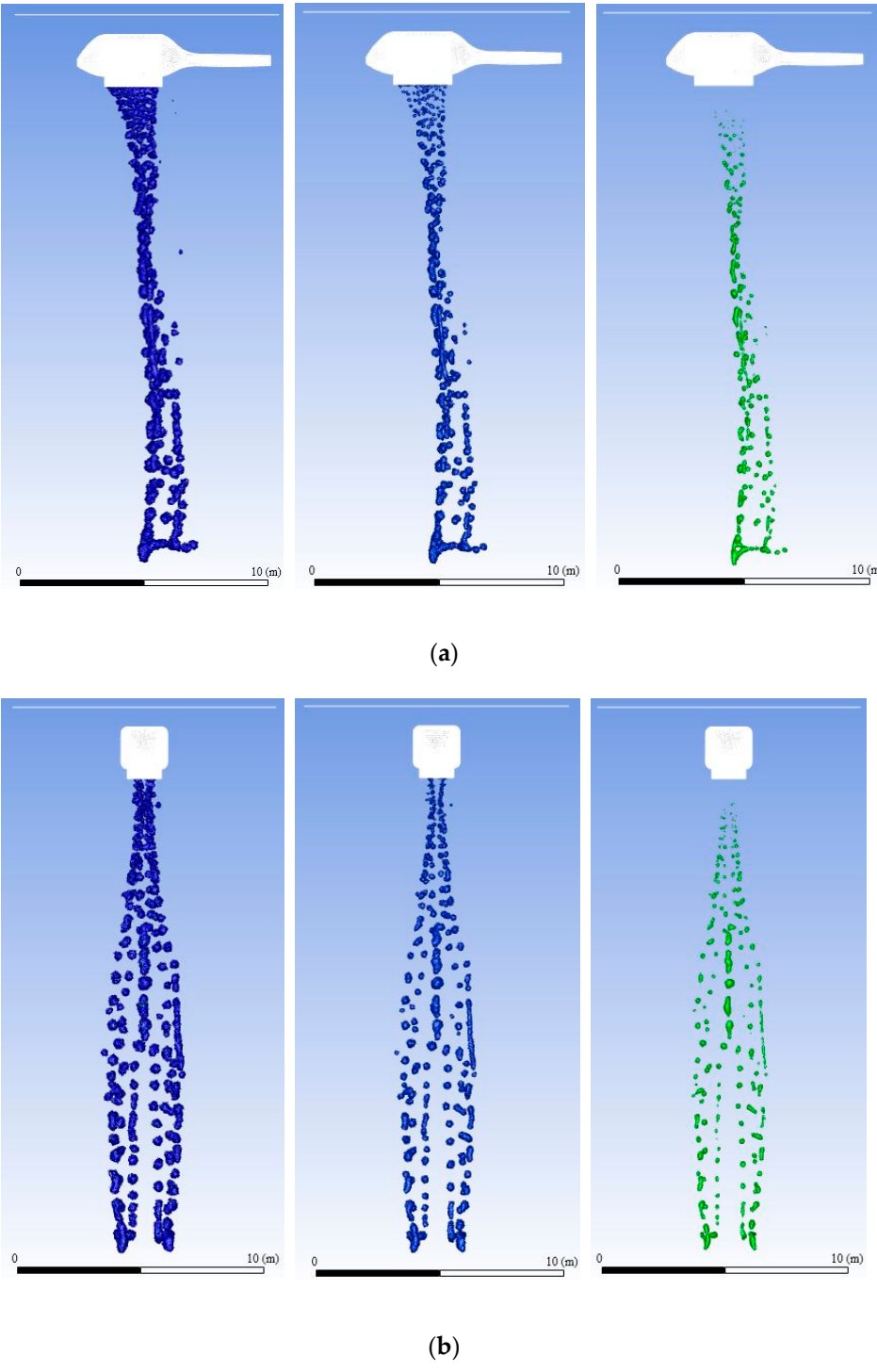

(**a**)

(**b**)

**Figure 19.** Contour surface of the two-phase, water–air volume fraction with fine meshes (the ground is at the ruler). (**a**) Contour surface and side view when the two-phase, water–air volume fraction was 0.01, 0.1, and 0.5, respectively, at *t* = 1.9 s. (**b**) Contour surface and front view when the two-phase, water–air volume fraction was 0.01, 0.1, and 0.5, respectively, at *t* = 1.9 s.

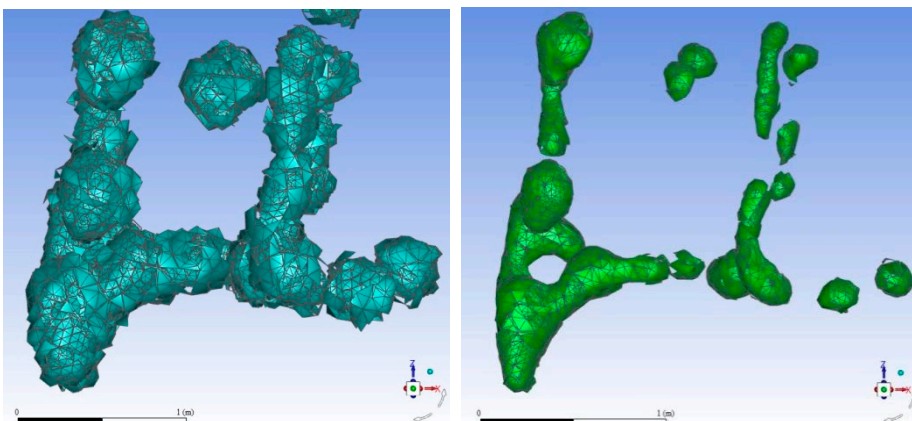

**Figure 20.** Contour surface of volume fraction near the ground and mesh cells in this region (the volume fraction was 0.01 in the (**left**) figure and 0.5 in the (**right**) figure).

It was calculated in the above example that, at $t$ = 1.9 s, the water was close to the ground. If the calculation had continued, it would quickly have become unstable, because in the very short time when the water masses or large droplets hit the ground, the water mass is deformed and broken, and bouncing motions and other complex deformation and motion phenomena occur. The current mesh cell size and time step cannot distinguish such rapid and drastic changes, leading to instability in the calculations. Assuming that the main flow characteristics and the water–air distribution were obtained when the water was close to the ground, and assuming that the water in the air fell to the ground according to its existing trajectory for the rest of its time airborne, the case that the water continued to fall to the ground was not considered in this paper. If the complex process of water hitting the ground is ignored and the ground is considered as a porous media model that allows only water to pass through, or if the ground is set as a pressure outlet boundary, the problem of unstable calculations can be avoided, and an approximate distribution of accumulated water on the ground can be obtained.

## 4. Simulation Conclusion Analysis and Rule of Water Dumping Out of Helicopter Belly Firefighting Tank

To provide theoretical guidance for helicopter firefighting practices, a study on parameters influencing the water dump distribution was performed in this section regarding two key parameters: the height of the tank bottom above the ground ($H$) and the wind speed ($U$). A total of 12 calculation examples were considered, with $H$ = 10 m, 20 m, and 30 m, and $U$ = 0 m/s, 5 m/s (Level 3), 10 m/s (Level 5), and 15 m/s (Level 7). The wind direction pointed to the rear of the helicopter, or in other words, the helicopter flew into the wind. The settings of the basic meshes, computational domain, mesh dissolution, and the calculation model and method in this section were the same as those used in the examples in Section 3.

### 4.1. Result Analysis and Summary of Initial Scenario before Water Dump

The influence of the height of the helicopter tank bottom above the ground and of wind speed on the distribution scope of wake flow was analyzed and is summarized as follows. Figure 21 shows the distribution of wake flow when the helicopter tank bottom was 10 m above the ground and the wind speed was 0 m/s, 5 m/s, 10 m/s, and 15 m/s. Figure 22 shows the distribution of wake flow when the helicopter tank bottom was 20 m above the ground, for the same values of wind speed. Figure 23 shows the distribution of wake flow when the helicopter tank bottom was 30 m above the ground, for the same values of wind speed. These figures show that wind speed had a very significant influence on the wake flow direction of the helicopter, and when the height above the ground was between 10 and 30 m, the wake flow could be deviated by nearly 30 degrees from the

straight rearward direction (the wind direction) by a wind speed of 5 m/s. The deviation increased to around 45 degrees at a wind speed of 10 m/s and around 60 degrees at a wind speed of 15 m/s. Wake flow was also influenced by the height above the ground: the higher the helicopter was above the ground, the lower the wake flow speed near the ground. When the helicopter hovered in a wind at 30 m/s, the wake flow speed near the ground was around 8 m/s. The ground in Figures 21–23 is near the ruler.

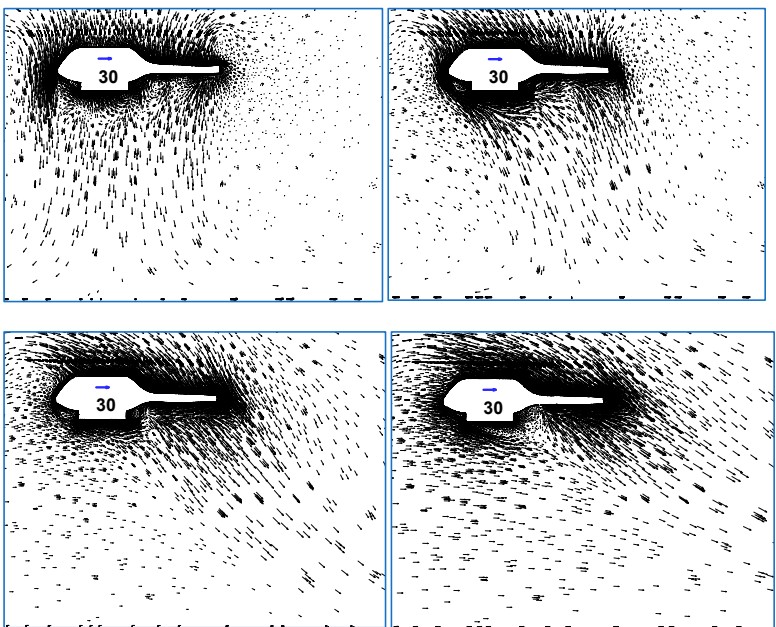

**Figure 21.** Distribution of wake flow when the helicopter tank bottom was 10 m above the ground (wind speed was 0 m/s, 5 m/s, 10 m/s, and 15 m/s; reference speed marked on the fuselage, indicated by a blue arrow, and the reference speed is 30 m/s).

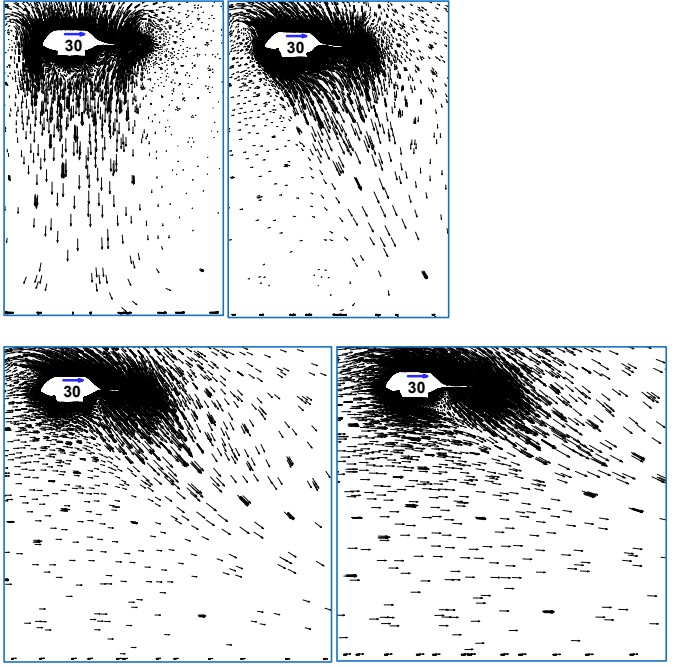

**Figure 22.** Distribution of wake flow when the helicopter tank bottom was 20 m above the ground (wind speed was 0 m/s, 5 m/s, 10 m/s, and 15 m/s; reference speed marked on the fuselage, indicated by a blue arrow, and the reference speed is 30 m/s).

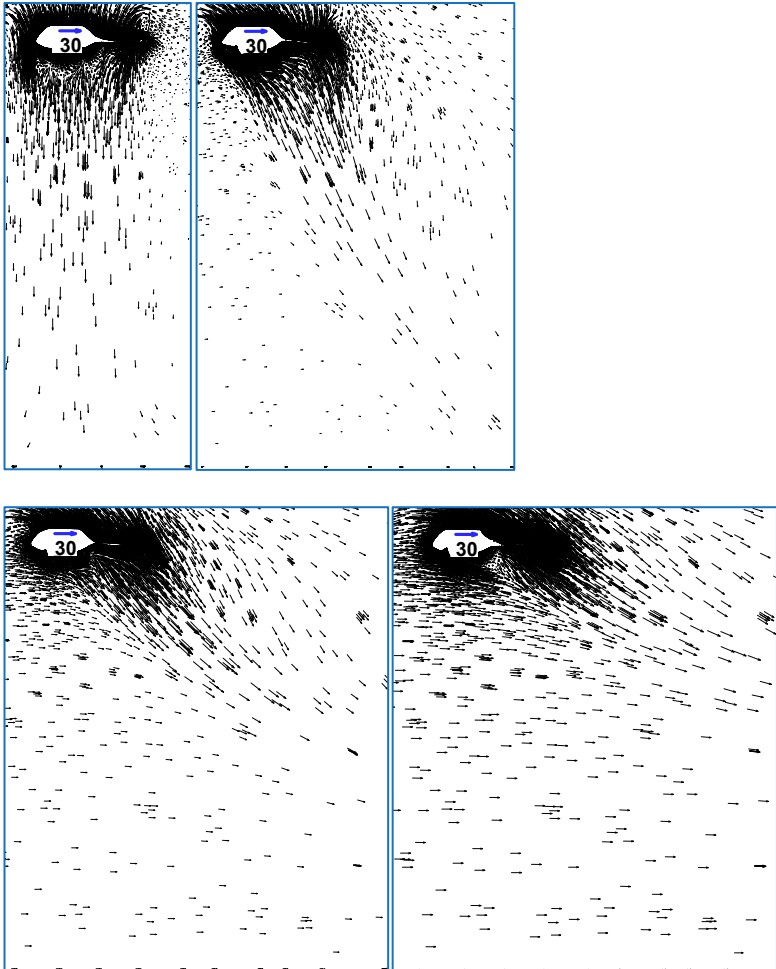

**Figure 23.** Distribution of wake flow when the helicopter tank bottom was 30 m above the ground (wind speed was 0 m/s, 5 m/s, 10 m/s, and 15 m/s; reference speed marked on the fuselage, indicated by a blue arrow, and the reference speed is 30 m/s).

*4.2. Transient Result Analysis and Summary of Water Dump*

Figure 24 shows the contour surface of the two-phase, water–air interface at a volume fraction of 0.01 when the helicopter tank bottom was 10 m above the ground and the wind speed was 0 m/s, 5 m/s, 10 m/s, and 15 m/s (*t* = 1.2 s, when the water mass was close to the ground). The figure shows that the motion trajectory of large water masses was basically not influenced at a wind speed of 5–10 m/s when the helicopter was 10 m above the ground, but at a wind speed of 15 m/s, the water masses moved slightly in the wind direction. The front view reveals that the water masses expanded significantly in the horizontal direction, with a horizontal expansion of nearly 50% compared with their size at a wind speed of 0 m/s.

Figure 25 shows the contour surface of the two-phase, water–air interface at a volume fraction of 0.01 when the helicopter tank bottom was 20 m above the ground and the wind speed was 0 m/s, 5 m/s, 10 m/s, and 15 m/s (*t* = 1.8 s, when the water mass was almost reaching the ground). Large water masses were basically not influenced at a wind speed of 5–10 m/s, but when the wind speed was 15 m/s, the water masses deviated significantly in the wind direction, with a deviation of 2–3 m compared with that at a wind speed of 0 m/s. The front view shows that the water masses also expanded significantly in the horizontal direction, with a horizontal expansion of around 7 m, which was around three times its horizontal extent at a wind speed of 0 m/s.

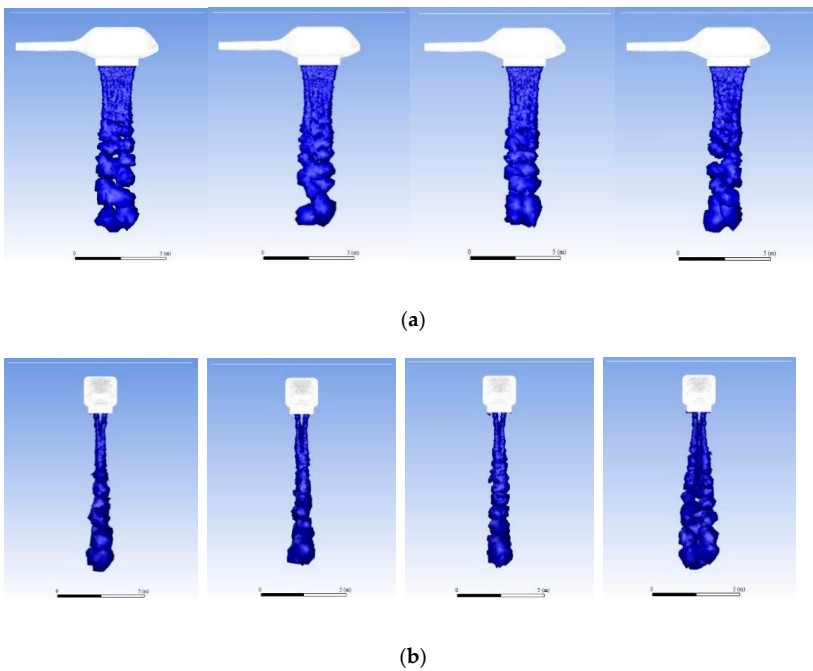

**Figure 24.** Contour surface of the two-phase, water–air interface at a volume fraction of 0.01 under different wind speeds (*H* = 10 m; the ground is at the ruler in the figure; *t* = 1.2 s). (**a**) Side view, showing *U* = 0 m/s, 5 m/s, 10 m/s, and 15 m/s, respectively, from left to right. (**b**) Front view, displaying *U* = 0 m/s, 5 m/s, 10 m/s, and 15 m/s, respectively, from left to right.

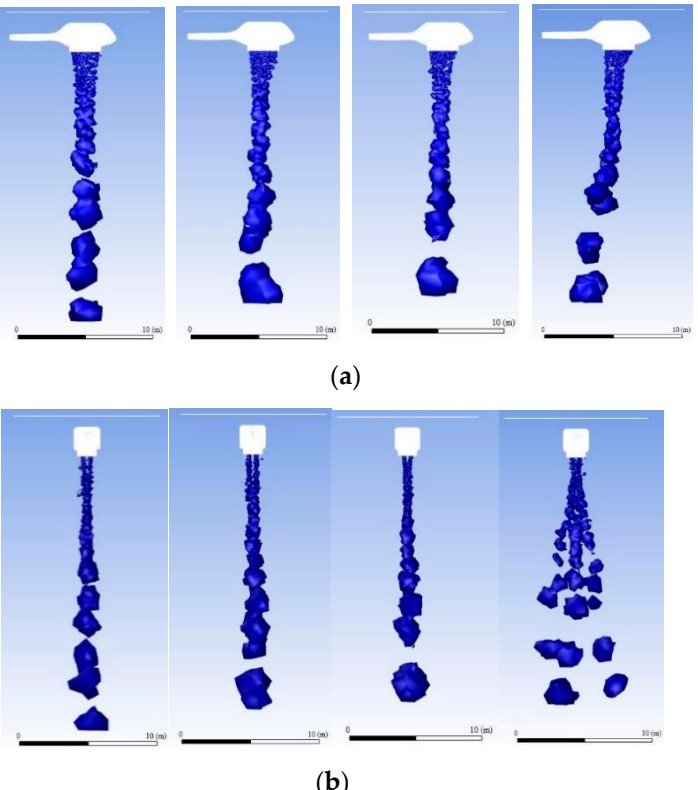

**Figure 25.** Contour surface of the two-phase, water–air interface at a volume fraction of 0.01 under different wind speeds (*H* = 20 m; the ground is at the ruler in the figure; *t* = 1.8 s). (**a**) Side view, displaying *U* = 0 m/s, 5 m/s, 10 m/s, and 15 m/s, respectively, from left to right. (**b**) Front view, showing *U* = 0 m/s, 5 m/s, 10 m/s, and 15 m/s, respectively, from left to right.

Figure 26 shows the contour surface of the two-phase, water–air interface at a volume fraction of 0.01 when the helicopter tank bottom was 30 m above the ground and the wind speed was 0 m/s, 5 m/s, 10 m/s, and 15 m/s, at around $t = 2$ s. In a similar manner, at a wind speed of 15 m/s, the water masses deviated significantly in the wind direction, and the water masses also expanded significantly in the horizontal direction, reaching around 9 m.

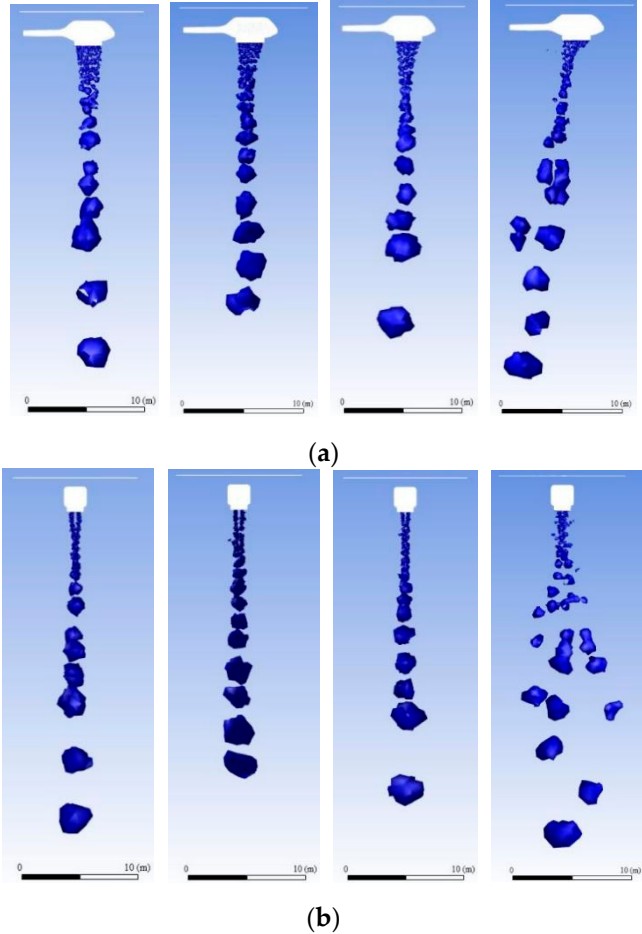

**Figure 26.** Contour surface of the two-phase, water–air interface at a volume fraction of 0.01 under different wind speeds ($H = 30$ m; the ground is at the ruler in the figure; $t = 2.1$ s). (**a**) Front view, showing $U = 0$ m/s, 5 m/s, 10 m/s, and 15 m/s, respectively, from left to right. (**b**) Front view, portraying $U = 0$ m/s, 5 m/s, 10 m/s, and 15 m/s, respectively, from left to right.

These observations inform us that wind speed had a very significant influence on the helicopter wake flow direction. When the height above the ground was 10–30 m, the wake flow could be deviated for nearly 30 degrees from the straight rearward direction (the wind direction) when the wind speed was 5 m/s. The deviation was around 45 degrees at a wind speed of 10 m/s and around 60 degrees at a wind speed of 15 m/s. The wake flow was also influenced by the height above the ground; the higher the helicopter, the lower the wake flow speed near the ground, and when the helicopter hovered at 30 m/s, the wake flow speed near the ground was around 8 m/s.

Large water masses were basically not influenced at wind speeds of 5–10 m/s, but when the wind speed was 15 m/s, the water masses deviated significantly in the wind direction and expanded significantly in the horizontal direction. The higher the water dump tank was above the ground, the greater the influence of wind speed on the water mass distribution.

*4.3. Forward Helicopter Flight*

In Section 4, it was assumed that the helicopter was hovering. In fact, before fire-fighting, the helicopter may fly forward at low speed above the targeted fire scene. For example, in the case where the forward flight speed was 10 m/s and there was no wind in the horizontal direction, this state could be approximated by the helicopter hovering in accordance with the principle of relativity of motion, with a headwind of 10 m/s. In addition, changes in the dip angle of the rotor discs were neglected in the calculation model. Regarding the problem of water dump distribution, although only the height of the helicopter tank above the ground and the wind speed were discussed in Section 4, the calculated result could be approximated by the circumstance where the helicopter was flying at a corresponding forward flight speed. The difference in the approximation came from changes in the dip angle of the rotor discs and the ground boundary layer, both of which had a small influence. Of course, if the hot air at the fire scene itself were taken into consideration, both wind speed and wind direction would have a significant influence on the hot air, and therefore the principle of relativity of motion might not be applicable. In the following discussion, the results obtained in Section 4 were applied to the circumstance of forward helicopter flight in accordance with the principle of relativity of motion.

Assuming that the helicopter tank was 20 m above the ground, the helicopter flew forward at a speed of 15 m/s (i.e., 54 km/h) and there was no wind (Figure 27b), then this would be equivalent to the case where the helicopter hovered against the wind at 15 m/s and the ground moved at 15 m/s in the wind direction (Figure 27a). Figure 27a shows that the helicopter hovered initially right above Point O on the ground, and when the headwind was 15 m/s, the water masses reached Point P on the ground at about 1.8 s, with a deviation of about 2–3 m from the wind direction. At this time, the water in the tank had run out, and most of the water dumped into the air would have fallen to the ground in approximately 1.8 s. Assuming that the ground moved leftward at a speed of 15 m/s, water fell finally at Point Q on the ground, and PQ was the region covered by most of the water, with an approximate length of 15 × 1.8 = 27 m. At this time, Point O had moved to Point O′.

Figure 27b shows that if the helicopter flew forward at a speed of 15 m/s, there was no wind, the ground was fixed, and initially the helicopter was right above Point O′ on the ground, it flew to be right above Point O on the ground 1.8 s later, the water dumped initially fell to Point P on the ground at this time. Similarly, around 1.8 s later, most of the water would have fallen, finally reaching Point Q on the ground. In this case, the PQ region was the region covered by most of the water, with an approximate length of 15 × 2 = 30 m. Figure 25 shows that the horizontal width of the region was around 7 m, the region covered an area of about 30 × 7 = 210 m$^2$, and after 1.12 m$^3$ of water were dumped, there would be about 5 mm of water accumulated per square meter in the covered region.

Figure 27b shows that if the intent is to dump water in the right region starting from Point P on the ground (PQ), the pilot needs to open the tank valve to dump the water when the helicopter is right above Point O′. The region covered by the water dumped depends on the height of the tank bottom above the ground, the flight speed, the wind speed, and the time required to empty the tank.

Note that in these calculations, foaming agents and other surface-active materials were not added to the water. If surface-active materials were added, formation of water droplets and the vaporization speed of water could be reduced, thus reducing the area covered by water on the ground and increasing the amount of water distributed per unit area.

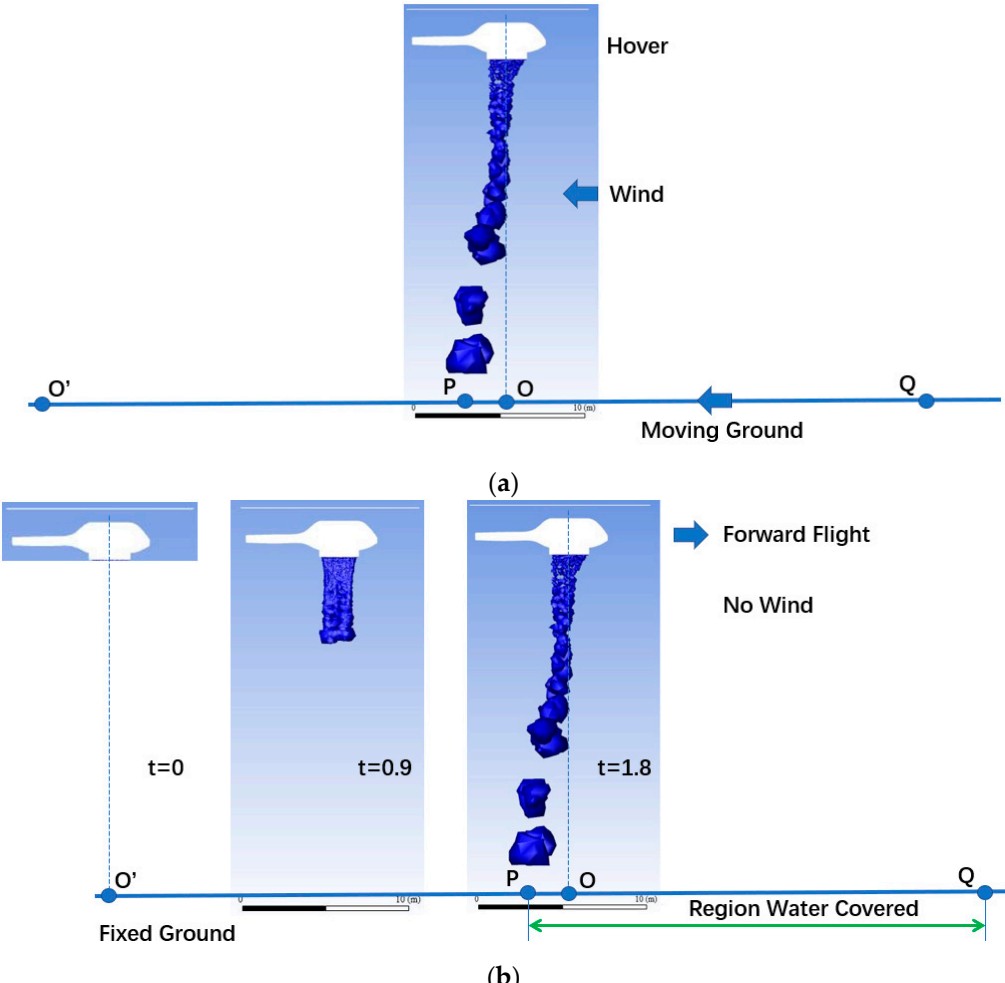

**Figure 27.** Water being dumped out of a helicopter tank and the distribution interval of water on the ground. (**a**) Helicopter hovering against the wind with the ground moving in the wind direction at wind speed. (**b**) Helicopter flying forward (forward flight speed = wind speed in a) with no wind and the ground fixed.

*4.4. Rule of Water Distribution on the Ground*

In accordance with the preceding data and calculation method, the region covered by water on the ground (also referred to as the water band in the literature) and average amount of water per unit area were given. Please note that, when calculating the average water amount per ground area, the accumulated water amount was considered in this paper instead of the water flow on the ground. However, in ANSYS FLUENT, the ground was set to have a non-slipping, fixed-wall condition, which, in this way, the water mass could flow all around after touching the ground; therefore, the accumulated water amount cannot be calculated automatically by this software. In order to calculate the accumulated water amount, the distribution of water mass just before touching the ground was taken into consideration in this paper, from which the water coverage area was given. Due to spatial–temporal changes in the water mass touching the ground, it is very complex to calculate the distribution of the water depth on the ground. Moreover, the breakup of drops and other phenomena were not taken into consideration in this paper; therefore, the average water depth was calculated in this paper only. Data on the hovering status of the helicopter, different wind speeds, the height of the tank bottom above the ground, the region covered by water on the ground, and the average amount of water were first provided, as shown in Table 1. The region covered by the water dump was approximated as a rectangle with area = longitudinal length × horizontal length. Table 1 shows that at wind speeds below 10 m/s,

the region covered by water on the ground was basically not influenced by the wind, but when the wind speed reached 15 m/s, the longitudinal dimension of the region covered by water on the ground decreased slightly, whereas the horizontal dimension increased greatly. Therefore, the area covered increased greatly, and under the premise of an unchanged total amount of water, the average water depth accumulated per unit area decreased greatly. The higher the tank bottom was above the ground, the greater the area covered by water on the ground, and the less the average water depth accumulated per unit area. However, the results at a height of 10 m were significantly smaller than those at 20 m and 30 m, although the results at the latter two heights were relatively close to each other.

**Table 1.** Influence of wind speed and height of the tank bottom above the ground on the region covered by water and the average water amount on the ground while the helicopter was hovering.

| Height of Tank bottom above the Ground/Wind Speed | 0 m/s | 5 m/s | 10 m/s | 15 m/s |
|---|---|---|---|---|
| 10 m | 2.4 m $\times$ 1.6 m = 3.84 m$^2$ 0.29 m water depth/m$^2$ | 2.2 m $\times$ 1.7 m = 3.74 m$^2$ 0.30 m water depth/m$^2$ | 2.1 m $\times$ 1.7 m = 3.57 m$^2$ 0.31 m water depth/m$^2$ | 2.1 m $\times$ 2.5 m = 5.25 m$^2$ 0.21 m water depth/m$^2$ |
| 20 m | 3.4 m $\times$ 3.0 m = 10.2 m$^2$ 0.11 m water depth/m$^2$ | 3.2 m $\times$ 3.0 m = 9.6 m$^2$ 0.12 m water depth/m$^2$ | 3.2 m $\times$ 3.2 m = 10.24 m$^2$ 0.11 m water depth/m$^2$ | 4.0 m $\times$ 7.0 m = 28.0 m$^2$ 0.04 m water depth/m$^2$ |
| 30 m | 2.9 m $\times$ 2.9 m = 8.41 m$^2$ 0.13 m water depth/m$^2$ | 3.0 m $\times$ 3.0 m = 9.0 m$^2$ 0.12 m water depth/m$^2$ | 3.0 m $\times$ 3.0 m = 9.0 m$^2$ 0.12 m water depth/m$^2$ | 5.0 m $\times$ 9.0 m = 45.0 m$^2$ 0.025 m water depth/m$^2$ |

As the helicopter flew forward, the region covered by water on the ground and the amount of water were also subject to the influence of forward flight speed. By the method discussed in Section 4.3, the average water distribution on the ground from the water dump when the helicopter flew forward (no wind) was as shown in Table 2. For forward flight, the higher the forward flight speed, the less the average water depth; a similar relation held for flight height. The average water depth per unit area was one order of magnitude less than in the cases of the corresponding hovering helicopter and wind speeds. For example, with a helicopter at a forward flight speed of 15 m/s and the tank bottom 30 m above the ground, the dumped water was distributed within a region of approximately 337.5 m$^2$, and the average water depth accumulated in this region per square meter was 0.3 cm. There were two working conditions having similar premises. The working condition of Hayden Biggs [15] was the most similar, with a forward flight speed of 70 km/h, a tank water-carrying capacity of 1.4 t, and a height above the ground of 24 m. The range of the water band on the ground was 120 m $\times$ 21 m, and the average water depth was 0.34 cm. The average water depth under similar working conditions in the present study was 0.3 cm, showing good agreement. Another similar working condition was that of Xie Yingmin [12]. In this case, the flight speed was 60 km/h (equivalent to 16.67 m/s), the height was 30 m, and the wind speed was 3 m/s. Their results showed that the maximum water belt on the ground was 110 $\times$ 25 m, the effective water belt was 65 $\times$ 15 m, and average water depth was 0.1–0.3 cm. The water-carrying capacity of the tank in the study by Xie Yingmin et al. [12] was 3 t, which was almost three times that in the present study, and assuming that the outlet flow was the same, its corresponding water belt area was also about three times that in this study. Therefore, the results of this study were basically consistent with the working conditions in Xie Yingmin et al. [12].

**Table 2.** Average water distribution on the ground with the helicopter flying forward when making a water dump (no wind).

| Height of the Tank bottom above the Ground/Forward Flight Speed | 5 m/s | 10 m/s | 15 m/s |
|---|---|---|---|
| 10 m | $5 \times 1.2 \times 1.7 = 10.2 \text{ m}^2$ 0.11 m water depth/$\text{m}^2$ | $10 \times 1.2 \times 1.7 = 20.4 \text{ m}^2$ 0.055 m water depth/$\text{m}^2$ | $15 \times 1.2 \times 2.5 = 45 \text{ m}^2$ 0.025 m water depth/$\text{m}^2$ |
| 20 m | $5 \times 2.0 \times 3.0 = 30 \text{ m}^2$ 0.037 m water depth/$\text{m}^2$ | $10 \times 2.0 \times 3.2 = 64 \text{ m}^2$ 0.0175 m water depth/$\text{m}^2$ | $15 \times 2.0 \times 7.0 = 210 \text{ m}^2$ 0.005 m water depth/$\text{m}^2$ |
| 30 m | $5 \times 2.5 \times 3.0 = 37.5 \text{ m}^2$ 0.03 m water depth/$\text{m}^2$ | $10 \times 2.5 \times 3.2 = 80 \text{ m}^2$ 0.014 m water depth/$\text{m}^2$ | $15 \times 2.5 \times 9.0 = 337.5 \text{ m}^2$ 0.003 m water depth/$\text{m}^2$ |

## 5. Conclusions

This study looked at two key parameters in firefighting helicopter operation: the height of the helicopter tank (H125/Isolair Eliminator II) above the ground (*H*) and the wind speed (*U*). After considering the relevant physical processes, such as the movement of water mass in the air, changes in the shape of the water mass, and the breakup of drops after water dump from the tank we decided to calculate the accumulated water amount on the ground. The height of the helicopter from the ground and the wind speed (or forward speed of the helicopter) are also important parameters influencing these physical processes, and hence were also considered in this work. The VOF model and adaptive mesh in ANSYS FLUENT were applied in this paper, which yielded the average water amount distributed on the ground after water dump from the helicopter. A study of the parameters influencing the water dump distribution was performed, considering *H* = 10 m, 20 m, and 30 m and *U* = 0 m/s, 5 m/s (Level 3), 10 m/s (Level 5), and 15 m/s (Level 7). The main conclusions were as follows:

(1) Wind speed had a significant influence on the direction of wake flow when the helicopter was hovering. The wake flow could by deviated by nearly 30, 45, and 60 degrees by wind speeds of 5 m/s, 10 m/s, and 15 m/s, respectively;

(2) The height above the ground also influenced the wake flow speed. The higher the helicopter was above the ground, the lower the wake flow speed near the ground. When the H125 helicopter hovered 30 m above the ground, the wake flow speed near the ground was about 8 m/s;

(3) There was no significant change in water mass distribution at wind speeds below 10 m/s, but when the wind speed rose to 15 m/s, the water masses deviated significantly in the wind direction and expanded significantly in the horizontal direction. Hence, the area covered by water on the ground increased significantly, and the average depth of water accumulated per unit area decreased significantly;

(4) The higher the tank was above the ground, the greater the area covered by water on the ground, and the less the average depth of water accumulated per unit area. However, the results at a height of 10 m are significantly less pronounced than those at 20 m and 30 m, although the results at the latter two heights are relatively close to each other;

(5) For forward flight, the higher the forward flight speed, the less the average depth of water on the ground; a similar relation held for flight height. The average depth of water was one order of magnitude less than in the cases of the corresponding hovering helicopter and the various wind speeds.

The following suggestions can be drawn from our results: if only the depth of water accumulated per unit area on the ground is considered when performing water dump firefighting, the helicopter should have the lowest possible forward flight speed and flight height and should perform firefighting under low wind speed conditions.

## 6. Future Work

The results calculated in this study show that the model developed here could be used to study the distribution of a water dump out of a helicopter tank, which has basically met the engineering requirements for firefighting with water dumps out of helicopters. To distinguish the water and air phase interfaces more meticulously and accurately, finer meshes and a smaller mesh adaptation cell scale would be required. In the meantime, in combination with "VOF-to-DPM" and the turbulent SAS model, small water droplets at mm level could be traced. For the fine meshes used in this study, the maximum mesh cell under the helicopter tank was 0.2 m in size, and the minimum mesh adaptation cell was 0.05 m, with total numbers of mesh cells up to 2 million. Using 84 CPU of the three calculation nodes of "Milky Way One", for example, it would take 17 h to calculate 1900 time steps (1.9 s). To distinguish water droplets at 0.01 m level (small water droplets at mm level required no distinction and were realized with the VOF-to-DPM model), the number of mesh cells would have to be increased by 125 times approximately, to 200–300 million mesh cells. If 840 CPU of "Milky Way One" were used, it would take about 10 days to calculate 1900 time steps, or about 200,000 CPU hours. Therefore, such calculation could be realized, but at an enormous cost.

This paper is merely a preliminary study on belly firefighting tank water dumping by helicopter and ground water distribution, and a coupling fire-field model must also be investigated. In the complete process of helicopter water dump firefighting, the wake flow of the rotor wings, the water–air flow and droplet dynamics, and combustion and the heat and smoke generated are all involved, making this still a very challenging problem of how to use the above models to establish an effective numerical calculation scheme.

**Author Contributions:** Conceptualization, J.L.; methodology, T.Z., S.L.; validation, T.Z. and C.W.; formal analysis, T.Z.; investigation, S.L.; data curation, T.Z.; writing—original draft preparation, T.Z.; writing—review and editing, T.Z. and S.L.; visualization, T.Z.; supervision, T.Z.; funding acquisition, J.L. All authors have read and agreed to the published version of the manuscript.

**Funding:** This work was funded by State Grid Corporation of China Science and Technology Project (5216A0210041) and the National Key Research and Development Plan (No. 2016YFC0800104).

**Institutional Review Board Statement:** Not applicable.

**Informed Consent Statement:** Not applicable.

**Data Availability Statement:** All data generated or analyzed are available on request through the author, Zhou Tejun, whose email address is zhoutejun1988@126.com.

**Acknowledgments:** The authors would like to thank the Editor and the reviewers for their comments and suggestions, which have been very helpful in improving the quality of this paper.

**Conflicts of Interest:** The authors declared that they have no conflicts of interest in this work.

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
