# Peer review of "Numerical Calculation and Analysis of Water Dump Distribution Out of the Belly Tanks of Firefighting Helicopters"

_safety, 2022_

Round 1

Reviewer 1 Report

TITLE: Seems to be OK.

ABSTRACT:

General comment: Seems to be OK.

KEYWORDS: OK.

INTRODUCTION:

General comment:

In general, well written and informative, although English phrasing needs improvement. I appreciate the enthusiasm of the Authors of being very informative but the Introduction is too large (discouraging) right now. A more synthetic approach would be beneficial for the paper.

Specific comments:

Lines 51-98, Table 1: compress the information in just one paragraph of 5 to 10 lines and use there as a background the information given in Table 1. Usually, not tables are to be included in introduction.

Lines 115-125: Need referencing. All the information coming in the following three paragraphs needs to be synthetized here in no more than 5-10 lines.

Lines 175-186: this belongs to materials and methods.

Lines 186-192: this belongs to discussion/conclusion.

Rest of the paper:

Seems to be particularly long. Perhaps the authors would try to shorten it to the most important features/things to be presented.

LITERATURE & CITING SYSTEM:

General comment:

Seems to be OK. Perhaps the inclusion of other references as commented above would enhance the paper.

Author Response

Response to the comments

Dear Prof. Zhou:

Thank you very much for your help in the review and revising of our manuscript. We deeply appreciate your valuable time and effect in improving our manuscript.

We’ve revised the manuscript based on the comments of all the reviewers. In the revised manuscript, the major changes are tracked. We’ve also prepared a comments and reply letter, in which all the questions raised by the reviewers were addressed. We hope that these documents could fulfill the requirements of the revision of manuscript and meet the standard of publication of the Journal of Fire Sciences.

Should you have any other questions on our manuscript, please let us know.

Thanks again.

Best regards

Sincerely yours,

TeJun Zhou

Reviewer(s)' Comments to Author:

  1. INTRODUCTION: General comment:

In general, well written and informative, although English phrasing needs improvement. I appreciate the enthusiasm of the Authors of being very informative but the Introduction is too large (discouraging) right now. A more synthetic approach would be beneficial for the paper.

Reply: The results of the experimental research work have been fully elaborated in Table 1. Therefore, we delete the detailed descriptions of helicopter models, operating conditions, and test results in the cited literature. In addition, we have added a discussion of the applicability and accuracy of physical models to the cited literature for numerical computation studies. Explanations of the physical models and computational schemes employed in this paper have been supplemented.

Please refer to page 1 to page 2 in the manuscript.

  1. Specific comments:

Lines 51-98, Table 1: compress the information in just one paragraph of 5 to 10 lines and use there as a background the information given in Table 1. Usually, not tables are to be included in introduction.

Lines 115-125: Need referencing. All the information coming in the following three paragraphs needs to be synthetized here in no more than 5-10 lines.

Lines 175-186: this belongs to materials and methods.

Lines 186-192: this belongs to discussion/conclusion.

Reply:

(1) Table 1 has been removed and described in the textï¼›Please refer to page 2 in the manuscript.

(2) All the information coming in the following three paragraphs were synthetized to no more than 10 lines. Please refer to line 78 to line 89 of page 2 in the manuscript.

(3) The content of Lines 175-186 in the original manuscript has been moved to Chapter 2 “Helicopter and tank models” in the revised manuscript. Please refer to line 101 to line 118 of page 3 in the manuscript.

(4) The content of Lines 186-192 in the original manuscript has been moved to Chapter 6 “Future work” in the revised manuscript. Please refer to line 101 to line 695 of page 700 in the manuscript.

Reviewer 2 Report

Overall, a very solid piece of work, but I felt the direct practical application was lacking. Perhaps the state of the art isn't there yet, but I would have liked to see direct recommendations in live fire-fighting operations as a function of fire size, fuel type, wind speed, and drop height. Even if that couldn't be provided until future work, a clear path to get there in the future work section would have been useful. I would also like to see an analysis of the how effective various helicopter drops are or are not for different fire types, but that is clearly beyond your current scope.

Author Response

Reviewer(s)' Comments to Author:

  1. Comments and Suggestions for Authors

Overall, a very solid piece of work, but I felt the direct practical application was lacking. Perhaps the state of the art isn't there yet, but I would have liked to see direct recommendations in live fire-fighting operations as a function of fire size, fuel type, wind speed, and drop height. Even if that couldn't be provided until future work, a clear path to get there in the future work section would have been useful. I would also like to see an analysis of the how effective various helicopter drops are or are not for different fire types, but that is clearly beyond your current scope.

Reply: The authors carried out experiments using buckets to fight different fires. The results named “Experiments in aerial firefighting with and without additives and its application to suppress wildfires near electrical transmission lines” of the trial were included in the 《journal of fire sciences》, but were not published.

This article excerpts the abstracts of the included articles. “A comparison scheme is proposed to extinguish non-uniform fire scenes and continuously uniform fire scenes using a helicopter’s bucket fire-extinguishing device to spray extinguishing agent. Pure water, and Class AB, gel, and Class A extinguishing agents were added to the bucket fire-extinguishing device to spray 4-layer, 6-layer, and 12-layer wood crib fires. It was discovered that the depth (the distance from the top of the wood crib) of effective cooling and prevention of temperature recovery by extinguishing agents was 0.36 m and that the cooling performance of the extinguishing agents in sequence from high to low was Class A extinguishing agent > gel extinguishing agent > Class AB extinguishing agent > pure water. Their capacity to prevent temperature recovery in the wood crib fires in sequence from high to low was gel extinguishing agent > Class A extinguishing agent > Class AB extinguishing agent > pure water. A device has been developed that can add extinguishing agent to the helicopter bucket efficiently, and its application on-site during the 2020–2021 Spring Festival and other events showed that it can quickly extinguish small-area wildfires near electrical transmission lines to reduce line trips due to wildfire.”

The next step will be to investigate the use of bucket fire-fighting operations as a function of fire size, fuel type, wind speed, and drop height.

Reviewer 3 Report

General concept comments:

This work presents a numerical study on water dumping from a helicopter. This may be helpful to study and establish guidelines on how the helicopters should be operated under different conditions.

While the paper is overall well written, some extra clarifications are needed to better understand the numerical study. Another important point is the number of references in the last 5 years. They are a small part of the overall bibliography

Detailed discussion:

The text that describes the domain is slightly confusing. I would like to see some of that information detailed in the Figures also. It would help to better understand the dimensions and characteristics of the simulated domain. I also think that images 2, 3, 4, and 5 are somewhat repetitive and a more “complex” image combining the different aspects necessary to construct the domain would give the reader a more interesting understanding of the domain. I do not believe that enough information on the domain is provided to allow the replication of the experiment.

Line 247-248: Why are these values used for the first example. Are there values that allow its validation?

Line 261-262: why these models/methods? Was there any validation of these methods? References?

Line 293-294: Again, why these values? Only the residuals were used to check the convergence of the solution. Was not possible to compare for example the pressure at boundary conditions?

It would make more sense to see section 3.3 before the discussion of the transient results.

Figure 20 and Figure 21: The reference to the wind velocity of each graph would be nice. I also think that more appropriate background and a more readable legend would improve the quality of the images.

Figure 23 is not easy to read.

Figure 24: This figure as well as the following could be improved. If the images were next to each other, it may be easier to infer the conclusions. 

Author Response

Response to the comments

Dear Prof. Zhou:

Thank you very much for your help in the review and revising of our manuscript. We deeply appreciate your valuable time and effect in improving our manuscript.

We’ve revised the manuscript based on the comments of all the reviewers. In the revised manuscript, the major changes are tracked. We’ve also prepared a comments and reply letter, in which all the questions raised by the reviewers were addressed. We hope that these documents could fulfill the requirements of the revision of manuscript and meet the standard of publication of the Journal of Fire Sciences.

Should you have any other questions on our manuscript, please let us know.

Thanks again.

Best regards

Sincerely yours,

TeJun Zhou

Reviewer(s)' Comments to Author:

  1. General concept comments:

This work presents a numerical study on water dumping from a helicopter. This may be helpful to study and establish guidelines on how the helicopters should be operated under different conditions.

While the paper is overall well written, some extra clarifications are needed to better understand the numerical study. Another important point is the number of references in the last 5 years. They are a small part of the overall bibliography

Reply: The reviewer made a good suggestion. The structure of the paper has been adjusted. Such as, the results of the experimental research work have been fully elaborated in Table 1. Therefore, we delete the detailed descriptions of helicopter models, operating conditions, and test results in the cited literature. In addition, we have added a discussion of the applicability and accuracy of physical models to the cited literature for numerical computation studies. Explanations of the physical models and computational schemes employed in this paper have been supplemented.

References have been updated with papers published in recent years.

  1. Detailed discussion:

The text that describes the domain is slightly confusing. I would like to see some of that information detailed in the Figures also. It would help to better understand the dimensions and characteristics of the simulated domain. I also think that images 2, 3, 4, and 5 are somewhat repetitive and a more “complex” image combining the different aspects necessary to construct the domain would give the reader a more interesting understanding of the domain. I do not believe that enough information on the domain is provided to allow the replication of the experiment.

Reply: The reviewer made a good suggestion.

Parts of Figures 2 to 5 are repeated. Figure 2 has both a helicopter model and a water tank model, but the water tank model is too small in the figure. Figures 3, 4, and 5 have been modified to make the tank model clear to the reader. Figure 3 shows the tank model alone, Figure 4 adds a description of the boundary conditions, and Figure 5 shows the base mesh for comparison with the dense mesh (Figure 6).

Please refer to from line 145 to 148 of page 4 in the manuscript.

Please refer to from line 194 to 197 of page 5 in the manuscript.

  1. Line 247-248: Why are these values used for the first example. Are there values that allow its validation?

Reply: The reviewer made a good suggestion. Because the height above the ground is moderate and the wind speed is high in this example, the trajectory and shape of the water mass change significantly in the air. Please refer to from line 157 to 162 of page 4 in the manuscript.

  1. Line 261-262: why these models/methods? Was there any validation of these methods? References?

Reply: The reviewer made a good suggestion. Since complex separation only existed near the helicopter and water tank, the wake flow with we were concerned under the water tank was not so complexed, and to which common turbulence models were applied; Please refer to from line 244 to 247 of page 7 in the manuscript.

  1. Line 293-294: Again, why these values? Only the residuals were used to check the convergence of the solution. Was not possible to compare for example the pressure at boundary conditions?

Reply: The reviewer made a good suggestion. corresponding to the rapidest water mass movement at approximately 1–2 cm. This ensured that there was sufficient time resolution, and the maximum number of iterations per time step was set to 50, so as to ensure residuals below 10-5. Please refer to from line 278 to 281 of page 9 in the manuscript.

  1. It would make more sense to see section 3.3 before the discussion of the transient results.

Reply: The reviewer made a good suggestion. Adjustments have been made as suggested by reviewers. Most of the original section 3.3 has been changed to section 3.1 mesh independence verification. Please refer to from line 168 of page 4 in the manuscript.

  1. Figure 20 and Figure 21: The reference to the wind velocity of each graph would be nice. I also think that more appropriate background and a more readable legend would improve the quality of the images.

Reply: The reviewer made a good suggestion. reference speed marked on the fuselage, indicated by a blue arrow, and The reference speed has been marked out. Please refer to from line 452 of page 16 to line 464 of page 17 in the manuscript.

  1. Figure 23 is not easy to read.

Reply: The reviewer made a good suggestion. The order of the sub-figures in this figure has been adjusted so that it can be read in comparison. Please refer to from line 490 to line 495 of page 18 in the manuscript.

  1. Figure 24: This figure as well as the following could be improved. If the images were next to each other, it may be easier to infer the conclusions.

Reply: The reviewer made a good suggestion. The order of the sub-figures in this figure has been adjusted so that it can be read in comparison. Please refer to from line 497 to line 500 of page 18 in the manuscript.

Reviewer 4 Report

This a good paper that studies the water dump distribution out of the belly tanks of firefighting helicopters by numerical analysis. The effects of the height of the tank above the ground and the wind speed on the wake flow and water distribution were discussed. The results suggested that the helicopter should have the lowest possible forward flight speed and flight height under low wind speed conditions when performing firefighting in a wildland fire. The results are valuable for wildland fire suppression. Besides, the authors should consider the influence of fire environment on the water distribution, including high air temperature and fire-induced flow.

Generally, the paper is well written, and the numerical method and the analysis are clearly described. I think this paper deserves to be published in Safety.

Author Response

Response to the comments

Dear Prof. Zhou:

Thank you very much for your help in the review and revising of our manuscript. We deeply appreciate your valuable time and effect in improving our manuscript.

We’ve revised the manuscript based on the comments of all the reviewers. In the revised manuscript, the major changes are tracked. We’ve also prepared a comments and reply letter, in which all the questions raised by the reviewers were addressed. We hope that these documents could fulfill the requirements of the revision of manuscript and meet the standard of publication of the Journal of Fire Sciences.

Should you have any other questions on our manuscript, please let us know.

Thanks again.

Best regards

Sincerely yours,

TeJun Zhou

Reviewer(s)' Comments to Author:

  1. Comments and Suggestions for Authors

This a good paper that studies the water dump distribution out of the belly tanks of firefighting helicopters by numerical analysis. The effects of the height of the tank above the ground and the wind speed on the wake flow and water distribution were discussed. The results suggested that the helicopter should have the lowest possible forward flight speed and flight height under low wind speed conditions when performing firefighting in a wildland fire. The results are valuable for wildland fire suppression. Besides, the authors should consider the influence of fire environment on the water distribution, including high air temperature and fire-induced flow.

Generally, the paper is well written, and the numerical method and the analysis are clearly described. I think this paper deserves to be published in Safety.

Reply: Thank you very much for your help in the review and revising of our manuscript. We deeply appreciate your valuable time and effect in improving our manuscript.

Round 2

Reviewer 1 Report

Dear Authors,

Thank you for the effort in improving the paper.

Now it looks much better.

Best regards,

R.

Reviewer 3 Report

All my concerns were addressed. Great Job. I recommend the publication of this work

This manuscript is a resubmission of an earlier submission. The following is a list of the peer review reports and author responses from that submission.